# POMAShiny: A user-friendly web-based workflow for metabolomics and proteomics data analysis

Pol Castellano-Escuder [1,2,3]*, Raúl González-Domínguez [1,3], Francesc Carmona-Pontaque[2,3], Cristina Andrés-Lacueva [1,3], Alex Sánchez-Pla [2,3]*

1 Biomarkers and Nutritional & Food Metabolomics Research Group, Department of Nutrition, Food Science and Gastronomy, Food Innovation Network (XIA), University of Barcelona, Barcelona, Spain, 2 Statistics and Bioinformatics Research Group, Department of Genetics, Microbiology and Statistics, University of Barcelona, Barcelona, Spain, 3 CIBERFES, Instituto de Salud Carlos III, Madrid, Spain

* pcastellano@ub.edu (PC-E); asanchez@ub.edu (AS-P)

**Data Availability Statement:** All source code and data files are available from the public GitHub repositories https://github.com/nutrimetabolomics/POMAShiny and https://github.com/

## Abstract

Metabolomics and proteomics, like other omics domains, usually face a data mining challenge in providing an understandable output to advance in biomarker discovery and precision medicine. Often, statistical analysis is one of the most difficult challenges and it is critical in the subsequent biological interpretation of the results. Because of this, combined with the computational programming skills needed for this type of analysis, several bioinformatic tools aimed at simplifying metabolomics and proteomics data analysis have emerged. However, sometimes the analysis is still limited to a few hidebound statistical methods and to data sets with limited flexibility. POMAShiny is a web-based tool that provides a structured, flexible and user-friendly workflow for the visualization, exploration and statistical analysis of metabolomics and proteomics data. This tool integrates several statistical methods, some of them widely used in other types of omics, and it is based on the POMA R/Bioconductor package, which increases the reproducibility and flexibility of analyses outside the web environment. POMAShiny and POMA are both freely available at https://github.com/nutrimetabolomics/POMAShiny and https://github.com/nutrimetabolomics/POMA, respectively.

## Author summary

Metabolomics and proteomics are two growing areas in human health and personalized medicine fields. Often, one of the main applications of metabolomics and proteomics is the discovery of novel biomarkers and new therapeutic targets in these areas. However, these data are extremely complex and hard to analyse, since they have a large number of features, several missing values, and often important clinical variables to consider in the analyses. Therefore, powerful and versatile tools are needed to provide efficient methods for data visualization and exploration, as well as a wide range of robust statistical methods to meet all data and users requirements. Although powerful tools do exist for the analysis

nutrimetabolomics/POMA. All source code and data files have also been provided as Supporting information files.

**Funding:** This work was completed as part of the European Joint Programming Initiative "A Healthy Diet for a Healthy Life" (JPI HDHL, http://www. healthydietforhealthylife.eu/) granted by MINECO (Spain, PCIN2017-076), and the ERA-Net cofound on Intestinal Microbiomics (ERAHDHL INTIMIC JPI HDHL) Project AC19/00096, AC19 00111, and CIBERFES funded by Instituto de Salud Carlos III and cofunded by European Regional Development Fund "A way to make Europe". CAL awarded by ICREA Academia 2018 and 2017SGR1546 grant from the Generalitat de Catalunya's Agency AGAUR. The funders had no role in study design, data collection and analysis, decision to publish, or preparation of the manuscript.

**Competing interests:** The authors have declared that no competing interests exist.

of these data, many of them are still limiting the analyses in terms of visualization and statistical analysis. To address this limitation and complement the existing tools, we have developed a web-based application, named POMAShiny, for the data analysis of metabolomics and proteomics. This novel and versatile tool offers a wholly interactive and easy-to-use environment for the analysis of these data, including numerous methods for pre-processing, data visualization and statistical analysis. The POMAShiny open-source tool is extremely flexible and portable, as it can be installed locally and freely accessed online at https://webapps.nutrimetabolomics.com/POMAShiny.

## Introduction

Metabolomics and proteomics are two rapidly growing areas of omics science that employ analytical techniques such as liquid chromatography (LC), gas chromatography (GC), capillary electrophoresis (CE), mass spectrometry (MS) and nuclear magnetic resonance (NMR) [1]. In turn, the results derived from these analytical techniques can be analysed in many different ways. Often, one of the main applications of metabolomics and proteomics is the characterization of novel therapeutic targets and patterns in human health and precision medicine fields [2, 3].

During the last decade, many open-source tools have emerged that contribute to the analysis of these complex data [4]. However, most of these tools remain very specific and may limit the analysis to a few statistical methods [5], forcing researchers to use an extensive battery of different tools to meet all the needs of the analysis.

Currently, statistical analysis of metabolomics and proteomics data is mainly conducted by using several programming tools [4] and/or via different web-based tools [6–9] according to the aims of the researchers. Web-based tools are often a very popular choice for researchers, as they provide fast and easy-to-use graphical interfaces that bring statistical analysis closer to the scientific community without the need for extensive programming skills. However, additional statistical approaches that are not implemented in these tools can be really useful in the data analysis process.

In an effort to contribute to the extension of available methods and options for metabolomics and proteomics data analysis, POMAShiny provides a comprehensive and structured workflow that covers the preprocessing, exploratory data analysis and statistical analysis of these data. This workflow is integrated into a user-friendly, attractive web-based user interface, mainly focused on statistical analysis. This new tool provides several powerful methods, including univariate statistical methods, multivariate and dimension reduction methods, feature selection methods, regularized regression analysis approaches, machine learning classification algorithms, prediction model strategies and several high-quality interactive visualization options.

This new tool is based on the POMA R/Bioconductor package (http://www.bioconductor. org). POMAShiny integrates many of the most widely used methods for metabolomics and proteomics data analysis [4, 5] and incorporates new useful and powerful alternatives.

The joint existence of both the POMA R/Bioconductor package and the POMAShiny web interface means a huge increase in the reproducibility of the tool, contributing to the reusability of previous existing methods in the R and Bioconductor environments [10, 11], as well as allowing easy extension, integration and interoperability with other workflows, such as the RforMassSpectrometry initiative (RforMassSpectrometry.org), which provides the data structures used by the POMA package. Therefore, users can perform the spectral data processing

and other routine MS workflow operations using the RforMassSpectrometry complementary packages, and then easily migrate to POMA to perform the statistical analysis without changing the data structure.

## Design and implementation

POMAShiny is a web-based tool wholly written in the open-source R programming language [10] and available under GPL-3.0 license. POMAShiny is powered by the Shiny framework [12] and all source code is available at the project's GitHub repository (https://github.com/nutrimetabolomics/POMAShiny).

On the one hand, POMAShiny's back-end structure uses the POMA [13], MSnbase [14] and tidyverse family [15] R packages to keep all code clean and as readable as possible, thereby facilitating the software maintenance. On the other hand, POMAShiny's front-end is based on the bs4Dash [16] R package, providing a highly easy-to-use dashboard design with most JavaScript features that makes the web interface very attractive for users. According to this design, the main menu with all panels and options is on the left side of the page while the main display screen is in the centre right of the page.

All functions provided in POMAShiny are tested with the testthat R package [17] on a continuous integration system using Travis, AppVeyor and GitHub Actions, covering tests on Linux, Mac and Windows with the current R versions and achieving more than 95% of code coverage [18] (https://github.com/nutrimetabolomics/POMA).

Users can download and launch POMAShiny locally (on Linux, Mac and Windows) or they can access the app online version hosted at https://webapps.nutrimetabolomics.com/POMAShiny. For a better experience, the authors recommend Safari or Chrome web browsers.

POMAShiny has been containerized using Docker. The Docker image is freely available at DockerHub, meaning a huge increase in the reproducibility, portability and scalability of the tool. Both local and Docker launch instructions are available at https://github.com/nutrimetabolomics/POMAShiny.

## Results

POMAShiny provides an analysis workflow structured in the four sequential and well-defined panels described below, namely 1) data upload, 2) preprocessing, 3) exploratory data analysis (EDA) and 4) statistical analysis, all of them with their respective subpanels (see Fig 1).

### Data upload

In order to keep "raw data" as "raw" as possible and create the ability to include covariates in the analysis, POMAShiny requires two comma-separated value (CSV) files as an input: the target (or metadata) file and the features file. The target file must provide the sample names in the first column and the group labels (e.g., control and case) in the second column. Optionally, from the third column (included), users can upload relevant covariates to be used in subsequent statistical analysis. The features file contains all quantified features in the experiment, one in each column starting from the first one. The order of rows in both files must be the same. Once these files are uploaded, POMAShiny converts them internally into an *MSnSet* object, as defined in the MSnbase R/Bioconductor package [14].

Once the target and features files are uploaded, users can select specific rows (samples) on the target file tab to create a data subset for those selected samples. If any selection is made, only the selected samples will be analysed.

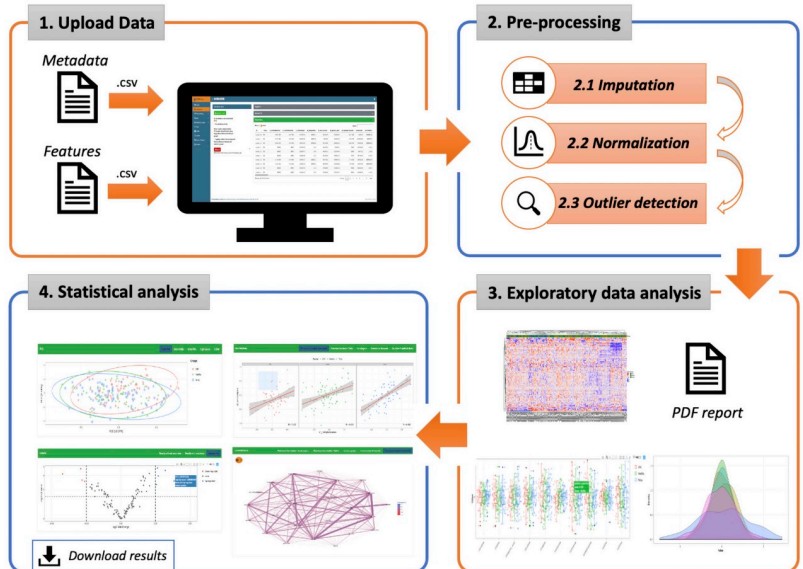

**Fig 1. POMAShiny's workflow.** The workflow is divided into four well-separated panels. Both target and features files are required as an input. Once these files are uploaded, the data are preprocessed and prepared for display in the exploratory data analysis panel (EDA). Finally, after the preprocessing and EDA panels, several statistical methods and options are provided in the statistical analysis panel, where users can analyse the data and download the results in both plot and table formats (*icons made by Freepik from* www.flaticon.com).

Furthermore, POMAShiny provides a function that allows users to combine different features that are part of the same entity. This optional operation can be very useful when the data contain different peptides that are part of the same protein or different ions representing the same compound. If users enable this option, a "grouping file" (CSV) indicating which features should be combined will be required. Several methods for performing this task are provided in POMAShiny, as well as the ability to download a table with the coefficients of variation of those combined features.

## Preprocessing

**Missing value imputation.** Often, for biological and technical reasons (e.g., inaccurate peak detection, values under the limit of quantification, etc.), some features cannot be identified or quantified in some metabolomics and proteomics samples [19, 20]. To address this problem, several methods have been developed and compared to identify the best approaches for data imputation in this context [19–21]. POMAShiny workflow offers a missing value imputation panel composed of different operations divided into three sequential steps: 1) distinction between zeros and missing values, 2) removal of features with a high percentage of missing values, and 3) imputation of remaining missing values.

First, if the data contain zeros or a combination of zeros and missing values, users can distinguish between them by using the algorithm provided by POMAShiny. For example, if the data contain both endogenous and exogenous compounds, the exogenous ones could be a real zero if they are not in the sample (absence), while the endogenous ones are unlikely to be real zeros (since they should always be in the sample, at least in very low concentrations). In that case, users could consider zeros as real zeros and impute only the missing values in the data. However, if users do not know the exact nature of the zeros in the data, or the difference

between zeros and missing values, the authors recommend considering all zeros as missing values.

Second, users are able to remove features of the data with more than a specific percentage of missing values in all the study groups (20% allowed by default). Finally, several imputation methods for dealing with the remaining missing values after the two previous steps are provided in POMAShiny. The available methods are the imputation by zero, the half minimum imputation, the imputation by median, the imputation by mean, the imputation by minimum and the *k*-nearest neighbours imputation (where missing values are imputed using the *k*-nearest neighbours algorithm [22]).

**Normalization.** It is generally accepted that some factors can introduce variability in metabolomics and proteomics data. Even if the data have been generated under identical experimental conditions, this introduced variability can have a critical influence on the final statistical results, making normalization a key step in the workflow [23]. These factors include: 1) differences in orders of magnitude between measured feature concentrations, 2) differences in the fold changes in feature concentration due to the induced variability, 3) large fluctuations in the concentration of some features under identical experimental conditions, 4) technical variability, and 5) heteroscedasticity [24]. POMAShiny provides six different normalization methods widely used in this field and compared in different studies [24, 25]. Here, the normalization process comprises both the transformation and scaling of the data in a single step. The methods available for this purpose are autoscaling, level scaling, log scaling, log transformation, vast scaling, and log pareto scaling.

**Outlier detection.** The last step provided in this panel is outlier detection and data cleaning. Outliers are defined as observations that are not concordant with those of the vast majority of the remaining data points [20]. Outliers in metabolomics and proteomics can be separated into biological and analytical outliers [26]. On the one hand, the first group reflects random and induced biological variations that make some observations different from others. On the other hand, the second group reflects different kinds of problems during the analytical process (e.g., sampling, storage) [26]. These values can have an enormous influence on the resultant statistical analysis, making it difficult to meet all required assumptions in the most commonly applied statistical tests as well as all required assumptions in many regression techniques and predictive modelling approaches. Therefore, outlier detection procedures are a critical point on which all subsequent analysis will depend (both inference and predictive statistics).

POMAShiny allows the analysis of outliers by different plots and tables as well as the possibility of removing statistical outliers from the analysis using different customizable parameters (see Fig 2).

Here, we propose an *ad hoc* multivariate outlier detection method based on the Euclidean distances among observations and their distances to each group centroid in a two-dimensional space (maximum, manhattan, canberra and minkowski distances are also available). Once the distances are computed, the classical univariate outlier detection formula $Q_3 + x * IQR$ is used to detect multivariate group-dependent outliers using the computed distances to each group centroid (*x*; the higher this value, the less sensitive the method is to outliers) (Fig 2).

## Exploratory data analysis

As discussed in the preprocessing section, many uncontrolled factors can introduce bias in a systematic manner in metabolomics and proteomics experiments: different chromatographic columns, eventual repair of the LC-MS system, different laboratory conditions, etc. [27]. Exploratory data analysis (EDA) can help in evidencing some of these confounding factors and possible outliers [27]. For that reason, it is highly recommended to perform an EDA

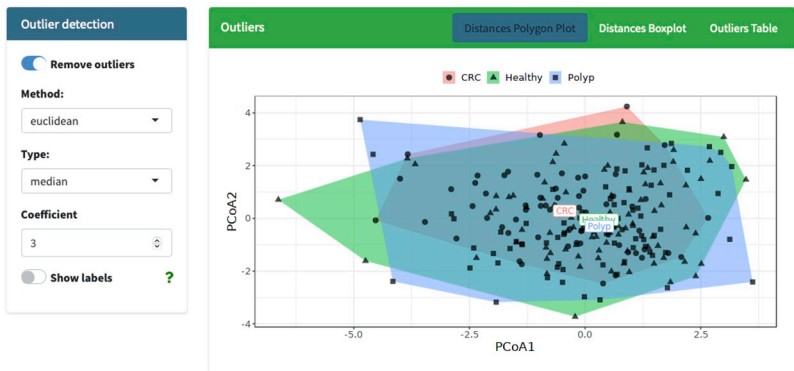

**Fig 2. Screenshot of the "Outlier detection" panel showing the Euclidean distances in principal coordinate space between samples and their respective group centroid.** ST000284 example data were used to create this plot.

before any statistical analysis [28, 29]. Moreover, in the case of negligible confounding factors or outliers, EDA can also be useful for getting a first idea of those most interesting features in the study.

POMAShiny offers several interactive and highly customizable plots designed to facilitate this process, providing a wide range of visualization options. The specific EDA functionalities implemented in POMAShiny are the volcano plot (for two-group studies), boxplots, density plot and heatmap. However, PCA (principal component analysis) and cluster analysis should also be considered in this section.

POMAShiny interactive boxplots are designed to visualize all features at once; however, users can easily customize this plot to display only features of interest. Unlike boxplots, the interactive density plot is designed to explore the distribution of the study groups. Alternatively, users can also display the distribution of specific features. POMAShiny also provides a clustered heatmap with a color stripe that corresponds to the group study label of each sample. Finally, for two-group studies, including a volcano plot in EDA can be very helpful for exploring those features that may be most influential in the study. POMAShiny's interactive volcano plot is based on the results of a T-test, where users can specify if the data are paired or if the variance is equal in the study groups.

**PDF report.** In an effort to facilitate the EDA process, POMAShiny includes a function to automatically generate a PDF report with a complete EDA, including all plots mentioned above. Users can generate the PDF report by clicking the "Exploratory report" button in the data upload panel.

The automatically generated PDF report provides information about the number of samples, features, covariates and the main study groups, as well as information on the percentage of total missing values in the data and the specific number of missing values per feature. Moreover, information on the number of zeros and features without variability is also provided. All the information provided in the PDF report is given in tables, plots and text format. A section with boxplots and density plots before and after missing value imputation and normalization (k-NN and log pareto scaling methods by default) is also included, providing users with valuable information about the preprocessing effect on the data. Furthermore, an outlier analysis, highly correlated features (r > 0.97), clustered heatmap and a PCA scores plot are also provided.

A POMAShiny PDF report helps users to have a quick and accurate description of the data. An example of this report is included as a vignette in the POMA package and it is also available at POMA's GitHub repository.

## Statistical analysis

This panel encompasses several statistical methods, from the most commonly used approaches in metabolomics and proteomics data analysis to other less frequently used methodologies in these fields. All statistical methods offered in POMAShiny (Table 1) are implemented in a highly user-friendly way and generate both downloadable tables and interactive plots as outputs.

**Univariate analysis.** POMAShiny offers four widely used methodologies for performing classical parametric and non-parametric univariate tests. On the one hand, T-test (two-group analysis) and ANOVA ($> 2$ group analysis) methods are available to perform parametric tests. In the ANOVA tab, an ANCOVA (analysis of covariance) model is also computed if covariates are included in the target file. On the other hand, the Mann-Whitney U test (two-group analysis) and Kruskal-Wallis test ($>2$ group analysis) are available for non-parametric analysis. Each of these methodologies offers customizable parameters to adjust the analysis to users and data requirements. Due to the large number of tests performed in these types of analyses, the FDR (false discovery rate) method is used to compute adjusted p-values.

**Limma.** Limma (linear models for microarray data) is a univariate method created for the statistical analysis of gene expression experiments as microarrays [30]. In recent years, this approach has become the main choice for many researchers to explore and identify differential expressed genes between two conditions. Due to the many similarities between metabolomics, proteomics and microarray data (often hundreds of features and small sample sizes, quantitative data, etc.), limma can be used in metabolomics and proteomics data sets when they meet the requirements (e.g., feature normal distributions). POMAShiny allows users to perform limma models easily and include covariates in the model, if necessary. If covariates are provided in the target file (e.g., batch effects, sex, age), two limma models—with and without covariates—are computed automatically, including the covariates in the model in the order in

**Table 1. Statistical methods provided in POMAShiny.** *Methods that allow the use of covariates.

| | | |
|---|---|---|
| Univariate methods | Parametric | T-test (paired/unpaired) |
| | | ANOVA |
| | | ANCOVA* |
| | | Limma* |
| | Non-parametric | Mann-Whitney U test (paired/unpaired) |
| | | Kruskal-Wallis |
| Multivariate methods | Unsupervised | PCA |
| | | k-means |
| | | Multidimensional scaling (MDS) |
| | Supervised | PLS-DA |
| | | sPLS-DA |
| Correlation methods | Parametric | Pearson's correlation* |
| | Non-parametric | Spearman's correlation* |
| | | Kendall's correlation* |
| | Visualization | Gaussian graphical models (GGMs) |
| Statistical learning methods | Regularized regression | LASSO regression |
| | | Ridge regression |
| | | Elasticnet regression |
| | Decision trees | Random forest |
| Generalized linear models | Logistic regression | Odds ratio calculation* |
| Permutation tests | Non-parametric | Rank products |

which they are provided in the target file (the further to the left, the more importance in the model). An interactive volcano plot based on the limma results is also generated in this tab.

**Multivariate analysis.** Multivariate methods focus analyses on the observation of more than one feature at a time, taking into account the different relationships between features. These methods can provide information about the structure of the data and different internal relationships that would not be observed with univariate statistics. However, the interpretation of these analyses can be more complex.

The most frequently used multivariate methodologies in metabolomics and proteomics statistical analysis are PCA, for unsupervised analysis, and PLS (partial least squares), for supervised analysis [31]. POMAShiny provides a collection of three different multivariate approaches powered by the mixOmics Bioconductor package [32]. The provided methods are PCA, PLS-DA (partial least squares discriminant analysis), and sPLS-DA (sparse partial least squares discriminant analysis).

PCA is an unsupervised method for dimension reduction that is done by calculating the data covariance matrix and performing eigenvalue decomposition on this covariance matrix without considering sample groups. In contrast, PLS-DA is a supervised method that uses the multiple linear regression method to find the direction of maximum covariance between the data and sample group [33]. sPLS-DA has been presented elsewhere [34] as an extension of sPLS (sparse partial least squares) [35] designed for classification problems. Note that while PCA is often used in exploratory data analysis, PLS-DA and sPLS-DA are used for classification and feature selection purposes, respectively. Several tuning parameters are available in all multivariate methods provided in POMAShiny. Users can define the number of components to compute, numerous graphical parameters, the number of features to select (in sPLS-DA), the VIP (variable importance in the projection) cut-off (in PLS-DA) and the cross-validation method to use, including both leave-one-out (LOO) and $k$-fold cross-validation.

**Cluster analysis.** Cluster analysis is also composed of multivariate methods, however this section is separated from multivariate methods to make POMAShiny structure clearer and more intuitive. The cluster analysis provided in POMAShiny allows users to explore different clusters in the data using the $k$-means algorithm [36]. $k$-means is an unsupervised method aimed at assigning all samples of the study to $k$ clusters based on the sample means. By default, the optimal number of clusters ($k$) is determined through the popular "elbow method". Alternatively, users can define a specific number of clusters.

To provide a multivariate visualization of computed clusters, POMAShiny projects these clusters in the first two dimensions of a multidimensional scaling (MDS) plot [37]. Many user-customizable parameters are offered to define the distance used in MDS calculation. POMA-Shiny provides a table with the assigned cluster to each sample and an interactive MDS plot with computed clusters. This feature serves the users both in terms of cluster analysis and in calculating a classic MDS, integrating two useful functionalities within the same tab. As mentioned before, this method can also be useful in EDA.

**Correlation analysis.** Correlation analysis is usually one of the preferred options for evaluating the strength of relationships between different features [38].

POMAShiny provides different approaches to conducting an accurate correlation analysis. First, POMAShiny provides a highly customizable and interactive scatterplot of pairwise correlation between features (Fig 3). Here, users can select two different features and explore them in a very comfortable way, as they are able to remove some points of the plot by clicking on them, drawing a smooth line based on a linear model, and exploring pairwise correlations within each study group and among factorial covariates (if they are provided). A downloadable table with all pairwise correlations between features is also provided. Moreover, POMAShiny provides a global correlation plot (or correlogram) and a network correlation plot. For all of

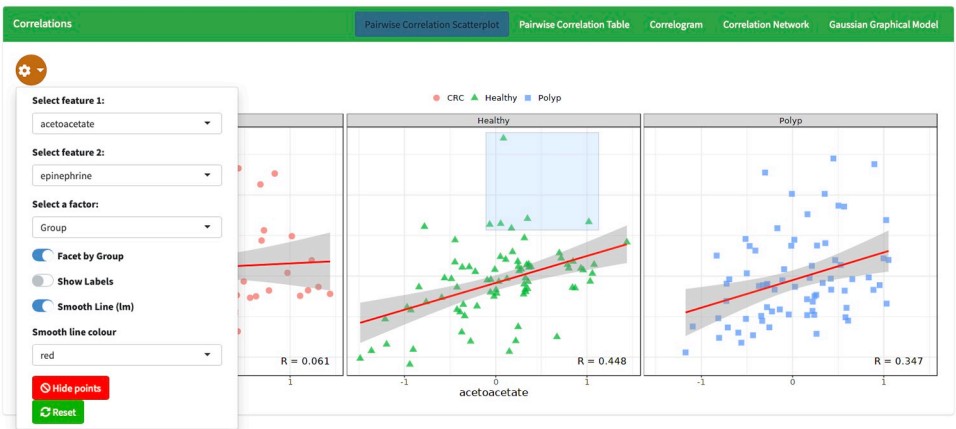

**Fig 3. Screenshot of the "Correlation Analysis" panel showing an interactive scatterplot of pairwise correlation between acetoacetate and epinephrine in the three different study groups "CRC", "Healthy" and "Polyp".** ST000284 example data were used to create these plots.

the above methods and options, the three most common methods for calculating correlation coefficients—Pearson, Spearman and Kendall—are available.

Lastly, POMAShiny provides an alternative method for correlation network visualization in this tab. Estimation of Gaussian graphical models (GGMs) through the glasso R package [39] is also provided here. Thus, users can define the regularization parameter to estimate a sparse inverse correlation matrix using LASSO [40] and visualize the resultant GGM in a network plot.

**Regularized regression.** Regularized regression is a type of regression that shrinks the coefficient estimates towards zero, providing less complex and flexible models but avoiding the risk of overfitting. POMAShiny offers three different regularization strategies: LASSO, ridge regression and elasticnet.

LASSO (least absolute shrinkage and selection operator) is a regression analysis method that sets some coefficients to zero, providing more compact and interpretable models [40, 41]. Because of that, LASSO is a very good approach for the statistical analysis of metabolomics and proteomics data, both in terms of feature selection and prediction model performance.

POMAShiny provides a function based on the glmnet R package [42] that allows users to create LASSO logistic regression models (two-group analysis) both for feature selection and prediction model purposes. If the purpose is not predictive, users can set the *test* set parameter to zero and the function will return interactive plots and tables referring to the LASSO model created using all samples of the study. Otherwise, if the purpose is to build a predictive model, users can select the proportion of samples that will be used as a *test* set. In the second case, POMAShiny will fit a LASSO model without using the *test* set and using it only to perform an external validation, providing users with numerous real prediction metrics (accuracy, accuracy confidence intervals, sensitivity, specificity, etc.). Alternatively, users can also perform elasticnet (defining a penalty parameter) and ridge regression models in the same tab. For all regularized regression strategies provided in POMAShiny, the lambda parameter is chosen automatically through internal *k*-fold cross-validation.

**Random forest.** In recent years, machine learning algorithms such as random forest have become very common in the analysis of omics data. These algorithms are constantly used both to rank the importance of features and to create prediction models. POMAShiny provides a classification random forest algorithm [43] designed for the creation of prediction models to

classify between two or more groups. This feature allows users to easily split data into *train* and *test* sets, where a *train* set is used to create the model and a *test* set is used only to perform an external validation. POMAShiny's random forest tab provides different tables and interactive plots with model metrics and the importance of features in the classification. In addition, the classification algorithm provided by POMAShiny returns the model confusion matrix and errors calculated using the *test* set, providing a real measure of model accuracy.

**Odds ratio.** Odds ratio (OR) calculation can be very helpful in visualizing and exploring the individual feature effects on the study outcome. POMAShiny includes an option to calculate OR based on a logistic regression model (two-group analysis). By changing the function parameters, users can easily define those features that will be included in the model and the ability to include study covariates in the model.

**Rank products.** The rank product is a statistical non-parametric test based on ranks of fold changes. This method has been used for several years to detect differentially expressed genes in microarray experiments [44]. However, in recent years this methodology has also become popular in other omics fields such as transcriptomics, metabolomics and proteomics [45]. POMAShiny includes an option to calculate rank products both for paired and unpaired samples with a set of customizable parameters. This function provides both tables and interactive plots showing the upregulated and downregulated features, respectively.

## Help and instructions

A comprehensive manual that details all POMAShiny functionalities is provided in the "Help" panel. In addition, users can also access all parameter-specific instructions by clicking on the "Help" icon available in each panel.

## Example data

POMAShiny includes two example data sets that are both freely available at https://www. metabolomicsworkbench.org. The example data set ST000284 consists of a targeted metabolomics three-group study and the example data set ST000336 consists of a targeted metabolomics two-group study. These two data sets allow users to explore all available functionalities in POMAShiny. Both data set documentations are available at https://github. com/nutrimetabolomics/POMA.

## Comparison with existing tools

Currently, most metabolomics and proteomics data analyses performed via web applications are conducted using the XCMS [6], MetaboAnalyst [7], Workflow4Metabolomics (W4M) [8] and Galaxy [9] tools. Among these tools, MetaboAnalyst and W4M are the most frequently used and complete in terms of statistical analysis [7].

Detailed comparisons among W4M, MetaboAnalyst and POMAShiny are exhibited in Table 2.

In terms of visualization and exploratory data analysis, only a few plots provided in MetaboAnalyst and W4M are interactive, while POMAShiny provides a whole interactive environment for almost all provided plots, offering a wide range of visualization options. As regards the importance of exploratory data analysis, POMAShiny dedicates a whole block of the workflow specifically to this issue, including an automatic PDF exploratory report. In contrast, both MetaboAnalyst and W4M provide an independent dendrogram plot, while in POMAShiny it is integrated into the heatmap.

As shown in Table 2, POMAShiny offers several methodologies for performing the three key preprocessing steps in metabolomics and proteomics: the missing value imputation,

**Table 2. Comparison of the main features of POMAShiny with Workflow4Metabolomics (W4M) and MetaboAnalyst web-based tools.** Symbols used for feature evaluations with "✓" for present and "✗" for absent.

| | Methods | POMAShiny | W4M | MetaboAnalyst |
|---|---|---|---|---|
| Visualization | Heatmap | ✓ | ✓ | ✓ |
| | Scatterplot (feature-feature) | ✓ | ✓ | ✗ |
| | Correlogram | ✓ | ✓ | ✓ |
| | Gaussian graphical models | ✓ | ✗ | ✗ |
| | Density plot (samples, features) | ✓ | ✗ | ✓ |
| | Boxplot (samples, features) | ✓ | ✗ | ✓ |
| | Volcano plot | ✓ | ✓ | ✓ |
| | Histogram | ✗ | ✓ | ✗ |
| | Dendrogram | ✗ | ✓ | ✓ |
| Preprocessing | Missing value imputation | ✓ | ✗ | ✓ |
| | Normalization | ✓ | ✓ | ✓ |
| | Outlier detection/cleaning | ✓ | ✗ | ✗ |
| Statistical analysis | T-test | ✓ | ✓ | ✓ |
| | ANOVA | ✓ | ✓ | ✓ |
| | ANCOVA | ✓ | ✗ | ✗ |
| | Limma | ✓ | ✗ | ✗ |
| | Mann-Whitney U test | ✓ | ✓ | ✓ |
| | Kruskal-Wallis | ✓ | ✓ | ✓ |
| | PCA | ✓ | ✓ | ✓ |
| | $k$-means | ✓ | ✗ | ✓ |
| | Multidimensional scaling | ✓ | ✗ | ✗ |
| | PLS-DA | ✓ | ✓ | ✓ |
| | OPLS(-DA) | ✗ | ✓ | ✓ |
| | sPLS-DA | ✓ | ✗ | ✓ |
| | Pearson's correlation | ✓ | ✓ | ✓ |
| | Spearman's correlation | ✓ | ✓ | ✓ |
| | Kendall's correlation | ✓ | ✗ | ✓ |
| | LASSO regression | ✓ | ✗ | ✗ |
| | Ridge regression | ✓ | ✗ | ✗ |
| | Elasticnet regression | ✓ | ✗ | ✗ |
| | Random forest | ✓ | ✓ | ✓ |
| | Support vector machine | ✗ | ✓ | ✓ |
| | Empirical bayesian analysis | ✗ | ✗ | ✓ |
| | Odds ratio calculation | ✓ | ✗ | ✗ |
| | Rank products | ✓ | ✗ | ✗ |

normalization and outlier detection. Being the implementation of a methodology for outlier detection and cleaning as part of preprocessing a significant improvement in the reproducibility of the results that other tools do not provide.

Overall, the primary strength of POMAShiny is the statistical analysis. Consequently, it is shown in Table 2 that POMAShiny provides the most commonly used statistical methods for metabolomics and proteomics data analysis and other very useful methods that the MetaboAnalyst and W4M tools do not provide (e.g., regularized regression methods, rank products), as well as the opportunity to include covariates in the analysis. The increasing complexity of experimental designs has made covariates such as sex and BMI (body mass index) have a high bias in the results. Thus, the ability to use statistical methods such as ANCOVA or limma—

which combine features data with other covariates—means an improvement in the accuracy and understanding of the results.

However, while POMAShiny does not offer some of the useful methods offered in MetaboAnalyst and W4M, such as orthogonal partial least squares discriminant analysis (OPLS-DA) or support vector machine (SVM), it does provide some useful methodological alternatives that these tools do not provide. These methodologies are LASSO, ridge regression, elasticnet regularization, ANCOVA, limma, rank products, odds ratio calculation and GGMs as a visualization option for correlations.

Finally, another significant advantage of POMAShiny is the predictive modelling strategy found in the regularized regression methods and in the random forest algorithm. POMAShiny allows users to easily create a random *test* set to perform an external validation of the model created with the *training* set, in contrast to MetaboAnalyst and W4M, which use the entire data set to create the models. This is a remarkable advantage as the results of POMAShiny both for regularized regression and random forest strategies provide real metrics of prediction models, allowing users to evaluate the model overfitting.

## Discussion

Despite the complexity of metabolomics and proteomics data, many of the most widely used web tools for statistical analysis of these data are not very versatile in terms of the input data structure and limit the analysis to a few statistical methods. POMAShiny is a web-based tool aimed at covering some of these data analysis bottlenecks, as it is a user-friendly and intuitive complementary tool that addresses some of the issues not covered by other tools. POMAShiny offers an integrated metabolomics and proteomics data analysis workflow with a wide range of possibilities both for data preprocessing and statistical analysis, including outlier detection methods, flexible exploratory data analysis operations, downloadable reports and several statistical methods, from simpler approaches such as univariate statistics to more complex methods such as regularization and machine learning prediction algorithms. This tool requires two files as an input—the target and features file—giving users the opportunity to include important study covariates in the analysis. This intuitive and powerful web interface allows users to perform an integrated data analysis in an interactive, well-documented and extremely user-friendly web environment, making the data analysis process more accessible to a wide range of researchers not so familiar with programming and/or statistical fields.

## Availability and future directions

The POMAShiny web application is hosted at our own server, https://webapps.nutrimetabolomics.com/POMAShiny, and is freely available to download at the project's GitHub repository, https://github.com/nutrimetabolomics/POMAShiny, where we also use the GitHub Issues tab as a discussion channel. Additionally, users can also download the POMAShiny Docker container from DockerHub. S1 Code contains source code, documentation and the Dockerfile of POMAShiny. S2 Code contains the POMA R/Bioconductor package source code and test data sets provided in POMAShiny.

POMAShiny is an open-source project that can be readily used and enhanced by the scientific community. Further expansion of the tool to cover more preprocessing and statistical methods will certainly increase its utility. The upcoming software enhancements will be directed at implementing new statistical methods, especially focused on machine learning algorithms, to enable more diverse and robust predictive abilities in the application. In addition, new visualization methods and different statistical analysis automatic reports will also be implemented.

## Supporting information

**S1 Code. Source code files and POMAShiny documentation.** In addition to the source code, the archive file contains the documentation for the installation and usage of the app and the Dockerfile to create a Docker image of POMAShiny.
(ZIP)

**S2 Code. Source code files and POMA Bioconductor package documentation.** In addition to the source code, the archive file contains the documentation of POMA and test data sets provided in POMAShiny.
(ZIP)

## Acknowledgments

We would like to acknowledge Dr Magalí Palau for her enthusiastic and valuable contribution at the start of this scientific development.

## Author Contributions

**Conceptualization:** Pol Castellano-Escuder, Raúl González-Domínguez, Francesc Carmona-Pontaque, Alex Sánchez-Pla.

**Funding acquisition:** Cristina Andrés-Lacueva.

**Software:** Pol Castellano-Escuder.

**Supervision:** Cristina Andrés-Lacueva, Alex Sánchez-Pla.

**Writing – original draft:** Pol Castellano-Escuder.

**Writing – review & editing:** Raúl González-Domínguez, Francesc Carmona-Pontaque, Cristina Andrés-Lacueva, Alex Sánchez-Pla.

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
