## [Decision Letter · Decision Letter 0]

23 Mar 2021

Dear Mr. Castellano-Escuder,

Thank you very much for submitting your manuscript "POMAShiny: an user-friendly web-based workflow for statistical analysis of mass spectrometry data" for consideration at PLOS Computational Biology.

As with all papers reviewed by the journal, your manuscript was reviewed by members of the editorial board and by several independent reviewers. In light of the reviews (below this email), we would like to invite the resubmission of a significantly-revised version that takes into account the reviewers' comments.

We cannot make any decision about publication until we have seen the revised manuscript and your response to the reviewers' comments. Your revised manuscript is also likely to be sent to reviewers for further evaluation.

Sincerely,

Manja Marz

Software Editor

PLOS Computational Biology

Manja Marz

Software Editor

PLOS Computational Biology

Reviewer's Responses to Questions

**Comments to the Authors:**

Reviewer #1: The authors present the POMAShiny R package/web application that bases on the

POMA Bioconductor package. The software does not implement new analytical

approaches but integrates a large set of commonly used analysis algorithms and

data visualization options and provides an easy-to-use web-based graphical user

interface enabling also the less experienced users to perform analyses. The

software is well written and also a docker image is available simplifying a

local installation which is crucial for users that are working with individual

level data and can not upload such data to public web services. While the

software is surely useful and well implemented the manuscript needs major

revisions. The content as well as the written english of the manuscript needs to

be improved.

Major issues

- The authors state that their software is designed for MS data analysis but the

data input format are simple tabulator delimited text files (same as one would

get from Microarray experiments or other quantitative large scale assays). The

tool does not support raw MS data handling or import nor does it provide any

methodology which is specific to MS data (such as combining feature

representing peptides of the same protein in proteomics or grouping of

features representing ions of the same compound for metabolomics). The authors

should either put less emphasis on MS data analysis and describe their tool as

a user interface for data analysis methods commonly used across all areas and

platforms used in computational biology, or should provide indeed MS specific

functionality.

- Also, the authors should consider to emphasize more the integration of

methodology provided by the POMA package which reduces the need for

transformation/converrsion between data types and objects.

- The authors need to compare their tool to other web-based tools such as iSee

or W4M.

- The written english of the document needs to be improved (see also minor

issues below).

- The authors state that POMAShiny allows more advanced users to replicate an

analysis in R (without the web frontend). This would require that the actual

commands used to perform the analysis are provided/shown to the user. Is this

the case? If not this argument should be removed.

- The authors should consider to provide the available algorithms/methods for

the statistical methods as a table and to emphasize more on what is available

in their software and less on the description on the various methods and why

they should be used. The main scope of the manuscript should be the

presentation of their software and less the description of data analysis

algorithms.

Minor issues

- The provided docker pull command does not work. Only "docker pull

pcastellanoescuder/pomashiny:release1.0.0" works. Authors should update/fix

that.

- lines 35-38: It is not clear how POMA integrates with packages from

RforMassSpectrometry as they mostly represent raw MS data (i.e. spectra data)

and POMA works exclusively on "feature tables" (i.e. quantified MS data). The

authors should clarify this.

- lines 44-46: it is not clear to the reviewer how the use of POMA and other

packages facilitates contribution of other users/developers. If there are

different contributors to the package/source code, the authors should provide

this information, otherwise I would suggest to remove/reformulate this

sentence.

- line 81: what is meant by "simplifying the indexation procedures among

data". The authors need to reformulate.

- Outlier detection: the authors should emphasize less on why outlier detection

is important and describe more/better how this can be achieved with their

tool.

- lines 330-331: The authors claim that their tool allows "...making data

analysis process more accessible to a wide range of researchers not so

familiar with programming and/or statistical fields." This might well be

correct, but especially for the target users (i.e. users less experienced with

the provided statistical methods) a more comprehensive tutorial and links to

descriptions of that methods is needed. Otherwise users can easily get mislead

by wrong results if inappropriate methods are used (or if they are used

incorrectly).

- Statistical analysis: the authors should include limma in the *Univariate

Analysis* section as linear model approaches are also univariate statistical

methods.

- The authors should detail if the linear model-based differential abundance

analysis (e.g. by limma) allows also to adjust for confounding

factors. Modeling e.g. batch effects or sex-age dependencies would be crucial

for many data analyses.

- Avoid the use of modulable.

- line 22: web -> web-based

- line 22-27: split the sentence. too complicated.

- line 45: the most readable -> as readable as possible

- line 119: what do the authors mean with "different modulable parameters".

- line 120-122: reformulate this sentence. it is not clear what "by default but

modulable" means. If euclidian distance is the default similarity measure but

other options are available too, list these.

- line 135: what do the authors mean with "modulable plots"? customizable plots?

- line 141-143: split this sentence into two.

- line 144: what do the authors mean with a "classical heatmap".

Reviewer #2: The authors have presented POMAShiny, a user-friendly web-based workflow for pre-processing and statistical analysis of mass spectrometry data. The manuscript is well written, and POMA work has been documented extensively. The source code and setup instructions are available at Github. Authors have covered many aspects of data pre-processing and implemented standard and commonly used ML analysis methods, exploiting R Shiney potential. Packages are presented in Biocunducor and containerised using Docker. The choices for packages/techniques used is excellent and up to date.

Some minor suggestion:

How would you compare the proposed set of web tool to other existing online resources? And What would distinguish it differently, and what would be its key advantages? Any limitation in using POMA or resources required?

Page 3 "To face this problem", better to use "to address this problem."

check code of conducts and remove/edit "[INSERT CONTACT METHOD]"

Regarding replacing a missing value with Zero. As you well know, there are many reasons for missing values, e.g., below LOQ or left-censored missing. If you use a determined value or zero, it may lead to certain biases, e.g., distortions of the distribution of missing variables and underestimations of the standard deviation. Therefore, I suggest making the caveats clear, and further explanations reference given to help guide the user to a better method choice, if possible.

In the Missing value section, are you considering "missingness" across entire data? If so, for example, if a sample has missing values beyond a certain percentage, would it be possible to remove that sample from the analysis?

Any plans for blanks contaminants subtraction and dealing with "data cleaning." as part of pre-processing

I would suggest differentiating between normalisation and scaling. In comparison, normalisation deals with the in samples variation, while scaling handles variations across variables (i.e. metabolites).

The quality of the figures needs improvement for publication. Finally, the help and online documentation can benefit from grammar and spelling revision to improve quality.

**Have all data underlying the figures and results presented in the manuscript been provided?**

Reviewer #1: Yes

Reviewer #2: Yes

PLOS authors have the option to publish the peer review history of their article (what does this mean?). If published, this will include your full peer review and any attached files.

Reviewer #1: No

Reviewer #2: **Yes: **Reza SALEK
---

## [Decision Letter · Decision Letter 1]

5 Jun 2021

Dear Mr. Castellano-Escuder,

We are pleased to inform you that your manuscript 'POMAShiny: a user-friendly web-based workflow for metabolomics and proteomics data analysis' has been provisionally accepted for publication in PLOS Computational Biology.

Best regards,

Manja Marz

Software Editor

PLOS Computational Biology

Manja Marz

Software Editor

PLOS Computational Biology

Reviewer's Responses to Questions

**Comments to the Authors:**

Reviewer #1: The authors successfully addressed all (of the many) major and minor points I

raised. The restructuring and editing largely improved the readability of the

manuscript. I particularly thank the authors for the extensive replies to each

issue as well as pointing to the related changes within the main text. Also the

new tables summarizing the available methods and the new section (and table)

with the comparison to other web-based tools are nice additions.

Reviewer #2: Thank you very much for the changes made. The comparison table looks great and certainly would be very valuable to the readership. I suggested changing the images format from Tiff (variable-resolution bitmap) to Vector based format to retain a much better quality images (e.g. PNG, SVG, ESP).

**Have the authors made all data and (if applicable) computational code underlying the findings in their manuscript fully available?**

Reviewer #1: Yes

Reviewer #2: Yes

PLOS authors have the option to publish the peer review history of their article (what does this mean?). If published, this will include your full peer review and any attached files.

Reviewer #1: **Yes: **Johannes Rainer

Reviewer #2: **Yes: **Reza M Salek

---

## [Editor Report · Acceptance letter]

23 Jun 2021

PCOMPBIOL-D-21-00274R1 

POMAShiny: a user-friendly web-based workflow for metabolomics and proteomics data analysis

Dear Dr Castellano-Escuder,

I am pleased to inform you that your manuscript has been formally accepted for publication in PLOS Computational Biology. Your manuscript is now with our production department and you will be notified of the publication date in due course.

With kind regards,

Katalin Szabo
